# A Rare Chromosome Rearrangement Leading to de la Chapelle Syndrome with a Mosaic 45,X Cell Line: (46,X,psu dic(X;Y)(p22.13;q11.221)/45,X/45,psu dic(X;Y)(p22.13;q11.221)

**DOI:** 10.3390/genes14010081

**Published:** 2022-12-27

**Authors:** Arthur Clement, Théo Dominot, Jeremy Chammas, Martine Montagnon, Marie Delcroix, Jérôme Pfeffer, Jean Michel Dupont, Aziza Lebbar, Patrice Clement, François Vialard

**Affiliations:** 1Clément Laboratory, F-93110 Paris, France; 2Department of Cytogenetics, APHP Centre, Université de Paris, Hôpital Cochin, F-75014 Paris, France; 3ZTP Laboratory, F-93170 Bobigny, France; 4Department of Genetics, CHI de Poissy St Germain en Laye, F-78300 Poissy, France; 5UMR-BREED, INRAE, ENVA, UVSQ, UFR SVS, Montigny le Bretonneux, F-78180 Saint-Quentin, France

**Keywords:** 46,XX *SRY*-positive male syndrome, chromosomal rearrangement, short stature, sex chromosome mosaic

## Abstract

Infertility affects about 15% of couples of childbearing age. About half of these cases can be attributed predominantly to a male factor, such as a quantitative or qualitative impairment in spermatogenesis. The first-line genetic screening for non-obstructive azoospermia is limited to karyotyping (to identify chromosome abnormalities) and Y chromosome microdeletions screening, with a view to explaining the spermatogenetic failure and evaluating the likelihood of sperm retrieval in a testicular biopsy. For patients with de la Chapelle syndrome (a 46,XX karyotype with the presence of *SRY* (Sex determining region Y) gene) and/or Y chromosome microdeletions, or sex chromosome mosaicism, sperm retrieval is usually unsuccessful. Here, we report a patient with de la Chapelle syndrome and a short stature caused by mosaicism and a very rare chromosome rearrangement: mos 46,X,psu dic(X;Y)/45,X/45,psu dic(X;Y). This case indicates that in de la Chapelle syndrome, X- and Y-chromosome breakpoint variability is high.

## 1. Introduction

The World Health Organization considers infertility (defined as the inability to conceive after 12 months or more of regular, unprotected sexual intercourse) to be a major health concern. It affects about 15% of couples of childbearing age. In about half of couples with infertility, the condition is of male origin [1]. Due to the prevalence of 47,XXY aneuploidy (resulting in Klinefelter syndrome) [2], karyotyping is the first-line genetic screening technique for infertile patients. However, many studies have reported other types of chromosome rearrangement [3]. It appears that reciprocal translocations involving only autosomes are more frequent in oligozoospermia, and that those involving gonosomes are rather associated with severe male infertility and azoospermia [4]. Furthermore, lower sperm counts are associated with a higher frequency of chromosomal anomalies; this is mainly due to Klinefelter syndrome, which is observed in 15% of men with azoospermia. Karyotyping studies have shown that Y chromosome rearrangements, especially those involving long arm deletions, are associated with infertility. It was suggested that these conditions were due to an azoospermia factor (AZF), of which three regions (AZFa, AZFb and AZFc) were subsequently identified [5]. Therefore, the first-line analysis in cases of severe oligospermia and azoospermia is based on karyotyping and Y chromosome microdeletion screening [3,6]. The results of these genetic analyses might rule out the use of assisted reproductive technology procedures in cases with (i) AZFa and/or AZFb microdeletions, leading, respectively, to Sertoli-cell-only syndrome and sperm maturation arrest [7], and (ii) a “46,XX” karyotype in a male individual.

The 46,XX male syndrome (also known as de la Chapelle syndrome) was first described in 1964 [8]. This rare disorder affects approximately one in 20,000 male infants [9,10]. As there are three phenotypes (depending on the development of male genitalia), the entity was subsequently renamed as “46,XX testicular disorder of sex development” [11]. The entity is due to an unbalanced translocation between the X and Y chromosome [12], which means that 80 to 90% of the individuals have Y chromosomal material, particularly the sex-determining region of the Y chromosome (*SRY*) gene [13]. The karyotype is, therefore, 46,X,der(X)t(X;Y)(p22.3;p11.2), with evidence of high breakpoint variability for the X and Y chromosomes [14]. All 46,XX men are sterile (due to azoospermia) and can present a variety of atypical sex development characteristics: abnormal hair distribution, gynecomastia, low testis volume, abnormal penis size, abnormal pubic hair development, erectile function disorders, and hypergonadotropic hypogonadism. All these characteristics are associated with the X chromosome disomy and the absence of most of the Y chromosome. Surprisingly, a significant proportion of “46,XX” men also have short stature, which might be associated with deletion of the *ARSE* gene (coding for arylsulfatase E) on the derivative X chromosome [14].

However, a testicular disorder of sex development can also be due to sex chromosome mosaicism with a monosomy X (45,X) cell line; this condition affects around one in 10,000 pregnancies. Furthermore, there is at least one cell line with a full or partial Y chromosome bearing the *SRY* gene. The clinical phenotypes range from typical female (usually considered to be Turner syndrome) to ambiguous genitalia (resulting in a diagnosis of mixed gonadal dysgenesis) and then typical male genitalia (probably the most common instance). Whatever the clinical phenotype, the health risks are similar to those found in Turner syndrome [15]. For men with typical male genitalia, fertility is related to the proportion of 46,XY cells in the testis.

Here, we report on a combination of de la Chapelle syndrome and sex chromosome mosaicism.

## 2. Case Report

The proband (a healthy man) consulted in our clinic for infertility. His height was 163 cm (minus 2 standard deviations), and he weighed 65 kg. The analysis of two consecutive semen samples revealed azoospermia. After counselling, the provision of consent and blood sampling, karyotyping showed a mosaic with three cell lines (46,X,der(X)/45,X/45,der(X): 70%, 25% and 5%, respectively; Figure 1a,b). We used fluorescent in situ hybridization with *SRY* probes (Figure 1c) and X and Y whole chromosome paints to characterize the derivative X chromosome (Figure 1d). Furthermore, Y chromosome microdeletion screening revealed AZFb and AZFc deletions.

To explore the rearrangement, a chromosome microarray (CMA) analysis (60 k oligonucleotide array; Agilent Technologies, Santa Clara, CA, USA) enabled us to determine the various chromosome breakpoints and led to the identification of a pseudodicentric chromosome: psu dic(X;Y)(p22.13;q11.221) (Figure 1e,f). The results for the patient and a normal male reference (ref: 46,XY) were arr[GRCh38]Xp22.33p22.13(1_16640171)x0[0.05],Xp22.13q28(18194098_155232904)x2[0.70], Yp11.32q11.221(1_18096418)x0[0.25],Yq11.221q12(16699218_28767604)x0,Yq12q12(59079666_59335913)x0[0.3]. The karyotype was mos 46,X,psu dic(X;Y)(p22.13;q11.221)[70]/45,X [25]/45,psu dic(X;Y)(p22.13;q11.221)[5]. Testicular sperm extraction was not indicated, and a clinical evaluation led to the couple being referred to a sperm donation program.

## 3. Discussion

We report here a rare case of sex chromosome mosaicism (with three cell lines) combined with a derivative X chromosome (leading to de la Chapelle syndrome). This event is rare; there are only three literature reports in males, with breakpoints on Xp22.3 and p11.32 (see Table 1) and a mosaic for two patients. The indication for karyotyping was Leri–Weill disease (LWD) or a sex development disorder. It is noteworthy that in contrast to the literature reports, the Y chromosome breakpoint in the present case was on the long arm on q11.221, which complicates the diagnosis with conventional cytogenetic analysis using G and R banding (Figure 1). Using only the SRY FISH probe, the karyotype could be misinterpreted as der(X)t(X;Y)(p22.3; p11.2). As we have reported previously, the present case also highlights the value of CMA analysis for better describing the chromosome rearrangement in this situation. Five literature cases have been reported in females, with the breakpoint on Yp11.2 in four cases (leading to *SRY* deletion). Only one case had a breakpoint on Yp11.32, but with a main clone 45,X accounted for 85%; this rearrangement was homogeneously transmitted to the male progeny (Wei et al., 2001).

Our observations confirm that Xp22.3 breakpoints are heterogeneous for the derived X chromosome in cases of de la Chapelle syndrome. The breakpoints are not all located close to the *PRKX* gene coding for protein kinase X (chrX: chrX:3522384–3631675), as previously reported [23,24]. This finding prompts us to hypothesize that non-allelic recombination between the homologous sequences of the short arms of the X- and Y-chromosomes is not the most likely mechanism: a nonrecombinant mechanism (like such nonhomologous end-joining) or a replication-based mechanism could also be considered. In patients with de la Chapelle syndrome, breakpoint sequencing will help to identify the most likely mechanism. In contrast, non-allelic recombination might be the main explanation for the dic(X;Y)(p22.33;q11.32), with recombination in the pseudo-autosomal region 1 and then *SHOX* gene deletion and LWD. This mechanism might also explain the idic(X)(p22.33) and all the isodicentric chromosomes.

Secondly, our observations confirm that the Y chromosome breakpoint can be on the short arm or the long arm of the Y chromosome; the latter situation would lead to a pseudodicentric chromosome, as reported previously [14]. This situation is quite rare because pseudodicentric X and Y chromosomes are usually pure derivative chromosomes, such as psu idic(Y) or psu idic(X) [25].

Regarding the mosaic monosomy X, the 45,X cell line has a full or partial Y chromosome bearing the *SRY* gene [15]. In the present case, we observed an association with a pseudodicentric chromosome with Y centromere inactivation (as shown by the karyotype). However, this pseudodicentric chromosome might have been lost during a cell division, producing the 45,X cell line. To the best of our knowledge, this has not been reported in the literature. It is noteworthy that the present case had a high proportion of cells with the *SRY* gene, which probably explains the normal male phenotype and the absence of ambiguous genitalia, even though there is sometimes a discrepancy between the cell proportion and the phenotype [26].

Unexpectedly, we observed the presence of a cell line with a pseudodicentric chromosome with a 16.64 Mb deletion. In the literature, an X chromosome with a large X chromosome deletion is never observed in males (where nullosomy for the deleted region is supposed to be lethal) and rarely in females (where these deletions are associated with a skewed X-chromosome inactivation pattern) [27]. This inactivation protects female carriers with a pathogenic X-deletion against severe clinical consequences. However, in the present case and considering the normal male phenotype, the psu dic(X;Y) was probably activated in the embryo during sex determination. The activation might have been cell-line-dependent and limited during embryo development, given (i) the typical positive selection for cells with a the psu dic(X;Y) inactivation and (ii) the skewed X-chromosome inactivation pattern for female carriers with a pathogenic X-deletion. Unfortunately, we were not able to study the X inactivation pattern in blood.

The patient’s short stature might be due to both the *SHOX* gene hemizygote deletion in the 45,X cell line and *ARSE* gene deletion in the cell lines with the derivative X chromosome [14]. Alternatively, the short stature might be due to *SHOX* and *ARSE* mutations; this is nevertheless rather unlikely and so was not investigated in the present study.

## 4. Conclusions

Our characterization of a very rare chromosome rearrangement confirmed the high variability of sex chromosome breakpoints in “46,XX,SRY-positive” patients and emphasized the complexity of chromosome rearrangements in humans.

## Figures and Tables

**Figure 1 genes-14-00081-f001:**
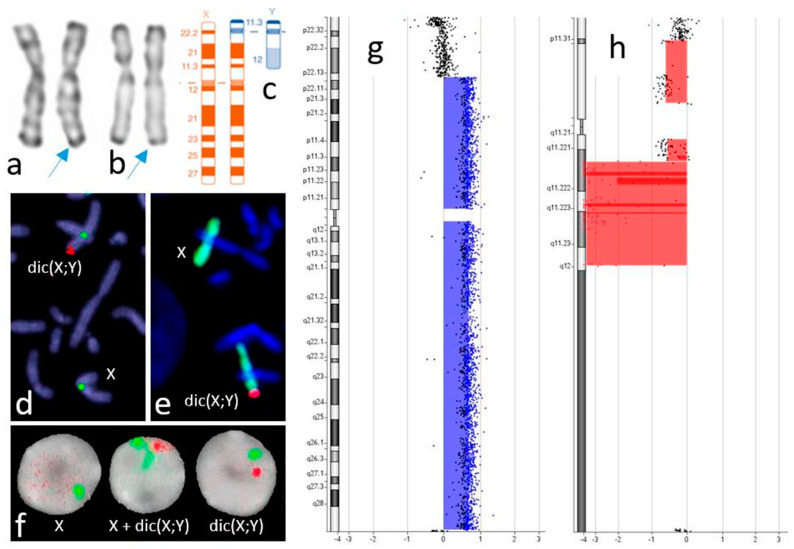
Cytogenetic analysis: the X and psu dic(X;Y) chromosomes using GTG (**a**) and RHG (**b**) banding, with the arrow showing the dic(X;Y); (**c**) ideograms for the X, dic(X;Y) and Y chromosomes; fluorescent in situ hybridization using *SRY* and X chromosome centromeric probes (**d**) and X and Y chromosome painting probes (**e**) on metaphase and (**f**) X and Y centromeric probes on nuclei; CMA results for the X (**g**) and Y (**h**) chromosomes, using a normal male reference.

**Table 1 genes-14-00081-t001:** Cases of idic(X;Y) reported in the literature.

	Cell Lines Number	Cell Line Proportion (%)		Chromosome Breakpoint		Phenotype	
		45,X	45,dic(X;Y)	X	Y	Sex	Other
Bernstein et al., 1987 [16]	1	0	0	q22	p11	Female	-
Baralle et al., 2000 [17]	1	0	0	p22.33	p11.2	Female	LWD
Wei et al., 2001 [18]	2	85	0	p22.33	p11.32	Female	LWD + TS
Burnside et al., 2008 [19]	1	0	0	p22.33	p11.2	Female	PD
Portnoï et al., 2012 [20]	2	75	0	p11.4	p11.2	Female	TS
Wei et al., 2001 [18]	1	0	0	p22.33	p11.32	Male	LWD
Mazen et al., 2013 [21]	3	23	12	p22.33	p11.32	Male	OT-DSD
Pavlistova et al., 2016 [22]	2	33	0	p22.33	p11.32	Male	LWD
Present case	3	25	5	p22.13	q11.221	Male	-

LWD: Leri–Weill disease; TS: Turner syndrome, PD: prenatal diagnosis; OT-DSD: ovotestis and disorder of sex development.

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
