# Peer review of "A Rare Chromosome Rearrangement Leading to de la Chapelle Syndrome with a Mosaic 45,X Cell Line: (46,X,psu dic(X;Y)(p22.13;q11.221)/45,X/45,psu dic(X;Y)(p22.13;q11.221)"

_genes, 2022, doi:10.3390/genes14010081_

Round 1
Reviewer 1 Report
Clement et al. summarized a case report on a male patient with de la Chapelle syndrome and sex chromosome mosaicism. The authors claimed that “in de la Chapelle syndrome, X- and Y-chromosome breakpoint variability is high”. To reach this conclusion, review and comparison with breakpoints of previously reported pseudodicentric (X;Y) rearrangements is required, yet the authors failed to do so. Further, presentation of the chromosome analysis result was sloppy and warrants considerable improvement. Please see below for details:
Major comments:
1. The “arylsulfatase E gene” was listed as a keyword, yet minimum discussion was included regarding this gene in the manuscript.
2. Data representation (Figure 1) was poor, including: 1) the banding resolution in Figure 1a was too low, thus difficult to appreciate the psu dic(X;Y) chromosome; 2) quality of banding in Figure 1b also low; 3) no indication of the colors of the FISH probes and chromosome painting probes in Figure 1c and 1d; 4) resolution of CMA results, particularly for the ideogram bandings were too low to read.
To improve the quality of Figure 1, consider adding an ideogram describing the rearrangement and properly label the FISH and chromosomal painting probe. Further, FISH images for all three cell lines should be included. The authors can use Figure 2 in Baralle et al 2000 (PMID 11186896) a reference.
3. The authors claimed that “non-allelic recombination between the homologous sequences of the short arms of the X- and Y-chromosomes is not the most likely mechanism” and proposed NHEJ/BIR mechanisms instead. Have the authors confirmed the absence of low-copy repeats at the breakpoint junctions of this case? Since the breakpoint junctions were not mapped to base pair resolution, this may be a farfetched statement.
4. Following the comment above, the authors should list previously published pseudodicentric (X;Y) rearrangements and compare the breakpoint junctions with the current case. This reviewer did a quick online query and found several reported patients:
PMID 22809487, 45,X/46,X,der(X)t(X;Y)(p11.4;p11.2);
PMID 18384143, a prenatal case with an X;Y translocation with the breakpoint at Xp22.3 and Yp11.2 resulting in a dicentric X;Y chromosome;
PMID 27588041, 46,X,dic(X;Y)(p22.33;p11.32)[20]/45,X[10] in an adult male;
PMID 11186896, a pseudodicentric derivative X;Y translocation chromosome with breakpoints at Xp22.3 and Yp11.2.
There are more cases out there; please include all reported ones. Such review and comparison could provide evidence for the authors’ statement on “in de la Chapelle syndrome, X- and Y-chromosome breakpoint variability is high”, as well as to propose rearrangement mechanism (NAHR or others).
Minor points:
1. Inconsistent fonts throughout the manuscript (e.g. Line 37, the sentence “It affects about 15% of couples of childbearing age” was in a different font; this is just one example).
2. Line 72 to 73, “… men have also short stature….” should be … men also have short stature.
3. Line 90, “A karyotype was proposed…” – was the karyotype (and CMA) performed using peripheral blood lymphocytes? Please state in the manuscript.
4. Line 93 to 94, “chromosome microdeletion screening revealed AZFb and AZFc deletions.” – did not see data from this screening in the manuscript.
5. Line 97 to 98, “CMA result for a normal male reference (ref: 46,XY)” should be CMA result using a normal male reference.
6. Line 98 to 99, nomenclature for “arr[GRCh38]Xp22.33p22.13(1-16640171)x0[0.05]“ should be “arr[GRCh38]Xp22.33p22.13(1_16640171)x0[0.05]” with underscore.
7. Resolution of the CMA (e.g. probe density and coverage regions) should be described.
Author Response
Firstly, my co-authors and I would like to thank you for giving us the opportunity to revise our manuscript. We also thank the reviewer for his/her help in improving the manuscript. We have considered his/her comments carefully, and provide point-by-point answers below.
Reviewer 1 :
Clement et al. summarized a case report on a male patient with de la Chapelle syndrome and sex chromosome mosaicism. The authors claimed that “in de la Chapelle syndrome, X- and Y-chromosome breakpoint variability is high”. To reach this conclusion, review and comparison with breakpoints of previously reported pseudodicentric (X;Y) rearrangements is required, yet the authors failed to do so. Further, presentation of the chromosome analysis result was sloppy and warrants considerable improvement. Please see below for details:
Major comments:
The “arylsulfatase E gene” was listed as a keyword, yet minimum discussion was included regarding this gene in the manuscript.
We have deleted that keyword.
Data representation (Figure 1) was poor, including: 1) the banding resolution in Figure 1a was too low, thus difficult to appreciate the psu dic(X;Y) chromosome; 2) quality of banding in Figure 1b also low; 3) no indication of the colors of the FISH probes and chromosome painting probes in Figure 1c and 1d; 4) resolution of CMA results, particularly for the ideogram bandings were too low to read. To improve the quality of Figure 1, consider adding an ideogram describing the rearrangement and properly label the FISH and chromosomal painting probe. Further, FISH images for all three cell lines should be included. The authors can use Figure 2 in Baralle et al 2000 (PMID 11186896) a reference.
The Figure has been improved, as requested.
The authors claimed that “non-allelic recombination between the homologous sequences of the short arms of the X- and Y-chromosomes is not the most likely mechanism” and proposed NHEJ/BIR mechanisms instead. Have the authors confirmed the absence of low-copy repeats at the breakpoint junctions of this case? Since the breakpoint junctions were not mapped to base pair resolution, this may be a farfetched statement.
In the literature, non-allelic homologous recombination (NAHR) has been mentioned as a rearrangement between PRKX and PRKY: this situation can be unambiguously ruled out here. The breakpoint on the Y chromosome is on Yp11.2 - a region considered to lack any X chromosome homology and/or X gene ancestry. Furthermore, our previous research (Capron et al.) showed that the breakpoint on the X chromosome is highly variable. In the present study, we reported a Xp22.13 breakpoint (far from the PRKX gene) and a breakpoint on Yq. A paragraph on this topic has been added (Lines 145 to 149).
Following the comment above, the authors should list previously published pseudodicentric (X;Y) rearrangements and compare the breakpoint junctions with the current case. This reviewer did a quick online query and found several reported patients: PMID 22809487, 45,X/46,X,der(X)t(X;Y)(p11.4;p11.2); PMID 18384143, a prenatal case with an X;Y translocation with the breakpoint at Xp22.3 and Yp11.2 resulting in a dicentric X;Y chromosome; PMID 27588041, 46,X,dic(X;Y)(p22.33;p11.32)[20]/45,X[10] in an adult male; PMID 11186896, a pseudodicentric derivative X;Y translocation chromosome with breakpoints at Xp22.3 and Yp11.2. There are more cases out there; please include all reported ones. Such review and comparison could provide evidence for the authors’ statement on “in de la Chapelle syndrome, X- and Y-chromosome breakpoint variability is high”, as well as to propose rearrangement mechanism (NAHR or others).
As requested, we have added the previously published rearrangements to Table 1. However, the breakpoints are clearly not similar: they are mainly on Yp11.2, which corresponds to X and Y chromosome fusion on Xpter and Ypter. The latter situation is also observed for idic(X)(p22.33). So, the situation is quite different for dic(X;Y)(p22.33;p11.32), where NAHR could be suspected and would (for example) explain the SHOX deletion on the dicentric chromosome. The karyotype in de la Chapelle syndrome (46,t(X;Y)(p22.3;p11.2)) is quite similar. Hence, we have revised the paragraph in accordance your remarks and the above information (lines 118 to 130).
Minor points:
Inconsistent fonts throughout the manuscript (e.g. Line 37, the sentence “It affects about 15% of couples of childbearing age” was in a different font; this is just one example).
We apologize for this unexpected error and have checked the font throughout the manuscript.
Line 72 to 73, “… men have also short stature….” should be … men also have short stature.
Done
Line 90, “A karyotype was proposed…” – was the karyotype (and CMA) performed using peripheral blood lymphocytes? Please state in the manuscript.
The sentences have been clarified (lines 92 and 97).
Line 93 to 94, “chromosome microdeletion screening revealed AZFb and AZFc deletions.” – did not see data from this screening in the manuscript.
We considered that this information was not interesting enough for inclusion in the Figure.
Line 97 to 98, “CMA result for a normal male reference (ref: 46,XY)” should be CMA result using a normal male reference.
Done.
Line 98 to 99, nomenclature for “arr[GRCh38]Xp22.33p22.13(1-16640171)x0[0.05]“ should be “arr[GRCh38]Xp22.33p22.13(1_16640171)x0[0.05]” with underscore.
Done - sorry for the typo.
Resolution of the CMA (e.g. probe density and coverage regions) should be described.
Done.
Reviewer 2:
The article is characterized by large and complex sentences. For the benefit of the reader and to make it enjoyable to read and easy to understand we would recommend shorter sentences. Especially in lines 61-64, 84-85, 114-117, 149-152.
Done.
Reviewer 3:
The MS presented a case with a rare chromosome rearrangement leading to de la Chapelle 2 syndrome. It is novelty and valuable for the understanding and investigation of male infertility. However,some improvements should be considered and revised .
Following comments regarding the manuscript are suggested.
- Controls and Methods are missing!How is this case diagnosed and characterized via figure 1 ? The methods and normal control sample should be provided and detailed.
With all due respect, we considered that showing a normal male karyotype and CMA results is of little interest. Regarding the FISH analysis, we have clarified things by adding the interphase results for the three cell lines. No metaphases were observed for the 45,der(X) line, and the 45,X FISH analysis is not interesting for inclusion in the Figure. We have therefore revised Figure 1 in accordance with reviewer 1’s request.
- Ethical approval and the questionnaire with informed consent also should be sated and provided.
Done, on line 99.
- Fonts and Formats should be careful inspected and formatted consistently.
We apologize for this unexpected error and have checked the font throughout the manuscript.
- Other theories or speculation, such as the patient’s short stature might be resulted from SHOX and ARSE muations. If the author can provide the related evidence, it will provide a great help to researchers in this field.
Done.
Reviewer 2 Report
The article is characterized by large and complex sentences. For the benefit of the reader and to make it enjoyable to read and easy to understand we would recommend shorter sentences. Especially in lines 61-64, 84-85, 114-117, 149-152.
Author Response

(The authors gave the same response as above.)

Reviewer 3 Report
The MS presented a case with a rare chromosome rearrangement leading to de la Chapelle 2 syndrome. It is novelty and valuable for the understanding and investigation of male infertility. However,some improvements should be considered and revised .
Following comments regarding the manuscript are suggested.
1. Controls and Methods are missing!How is this case diagnosed and characterized via figure 1 ? The methods and normal control sample shoud be provided detailedly.
2. Ethical approval and the questionnaire with informed consent also should be sated and provided.
3. Fonts and Formats should be careful inspected and formatted consistently.
4. Other theories or speculation, such as the patient’s short stature might be resulted from SHOX and ARSE muations. If the author can provide the related evidence, it will provide a great help to researchers in this field.
Author Response

(The authors gave the same response as above.)

Round 2
Reviewer 1 Report
The authors addressed all of this reviewers' comments in the revised manuscript, except for this one: "was the karyotype (and CMA) performed using peripheral blood lymphocytes?"
Please add this information in the manuscript.
Reviewer 2 Report
I enjoyed reading the manuscript and I congratulate the authors.